

# Differences in persistence between dogs and wolves in an unsolvable task in the absence of humans

Akshay Rao[1,2,*], Lara Bernasconi[1,3,*], Martina Lazzaroni[1,2], Sarah Marshall-Pescini[1,2] and Friederike Range[1,2]

[1] Wolf Science Center, Domestication Lab, Konrad Lorenz Institute of Ethology, Veterinärmedizinische Universität Wien, Vienna, Austria
[2] Comparative Cognition, Messerli Research Institute, Veterinärmedizinische Universität Wien, Vienna, Austria
[3] Department of Comparative Cognition, University of Neuchâtell, Neuchâtell, Switzerland
[*] These authors contributed equally to this work.

Corresponding author
Akshay Rao,
akshay.rao@vetmeduni.ac.at

## ABSTRACT

Despite being closely related, dogs perform worse than wolves in independent problem-solving tasks. These differences in problem-solving performance have been attributed to dogs' greater reliance on humans, who are usually present when problem-solving tasks are presented. However, more fundamental motivational factors or behavioural traits such as persistence, motor diversity and neophobia may also be responsible for differences in task performance. Hence, to better understand what drives the differences between dogs' and wolves' problem-solving performance, it is essential to test them in the absence of humans. Here, we tested equally raised and kept dogs and wolves with two unsolvable tasks, a commonly used paradigm to study problem-solving behaviour in these species. Differently from previous studies, we ensured no humans were present in the testing situation. We also ensured that the task was unsolvable from the start, which eliminated the possibility that specific manipulative behaviours were reinforced. This allowed us to measure both persistence and motor diversity more accurately. In line with previous studies, we found wolves to be more persistent than dogs. We also found motor diversity to be linked to persistence and persistence to be linked to contact latency. Finally, subjects were consistent in their performance between the two tasks. These results suggest that fundamental differences in motivation to interact with objects drive the differences in the performance of dogs and wolves in problem-solving tasks. Since correlates of problem-solving success, that is persistence, neophobia, and motor diversity are influenced by a species' ecology, our results support the socioecological hypothesis, which postulates that the different ecological niches of the two species (dogs have evolved to primarily be scavengers and thrive on and around human refuse, while wolves have evolved to primarily be group hunters and have a low hunting success rate) have, at least partly, shaped their behaviours.

## INTRODUCTION

Animals need to solve various ecological and social problems to survive. Studies across taxa have found problem-solving success to depend on several psychological propensities (referred to as the 'correlates of problem-solving success'). These include neophobia (the fear of new situations or objects), motor diversity and flexibility (the repertoire of problem-solving behaviours an animal displays, and its ability to find novel solutions to already known problems, or use known solutions to solve novel problems) and persistence (defined as task directed motivation and quantified as the amount of time an animal spends tackling a task) (*Lefebvre, Reader & Sol, 2004*; *Biondi, Bó & Vassallo, 2010*; *Hiestand, 2011*; *Cole, Cram & Quinn, 2011*; *Morand-Ferron et al., 2011*; *Thornton & Samson, 2012*; *Benson-Amram & Holekamp, 2012*; *Griffin & Guez, 2014*; *Moretti et al., 2015*; *Griffin & Diquelou, 2015*; *Huebner & Fichtel, 2015*; *Udell, 2015*; *Borrego & Gaines, 2016*). These correlates are interconnected: behavioural flexibility is positively correlated with persistence (*Morand-Ferron et al., 2011*; *Benson-Amram & Holekamp, 2012*; *Griffin & Guez, 2014*; *Huebner & Fichtel, 2015*; *Borrego & Gaines, 2016*) and both are negatively correlated with neophobia (*Bouchard, Goodyer & Lefebvre, 2007*; *Biondi, Bó & Vassallo, 2010*; *Thornton & Samson, 2012*; *Sol, Griffin & Bartomeus, 2012*; *Benson-Amram & Holekamp, 2012*; *Griffin & Guez, 2014*; *Moretti et al., 2015*; *Borrego & Gaines, 2016*). They are influenced by a species' ecology, social structure and living conditions (*Webster & Lefebvre, 2001*; *Lefebvre, Reader & Sol, 2004*; *Cauchard et al., 2013*; *Griffin, Diquelou & Perea, 2014*). For example, birds in variable environments and habitats were found to be less neophobic, have greater motor diversity and were more behaviourally flexible than conspecifics in more stable environments (*Mettke-Hofmann, Winkler & Leisler, 2002*; *Sol, Lefebvre & Rodriguez-Teijeiro, 2005*; *Sol et al., 2011*; *Kozlovsky, Branch & Pravosudov, 2015*). Persistence was higher in social carnivores than in closely related non-social ones, as well as in captive hyenas than in wild conspecifics (*Benson-Amram, Weldele & Holekamp, 2013*; *Borrego & Gaines, 2016*). An animal's personality (or behavioural type) has also been shown to play a role in problem-solving styles (*Sih & Del Giudice, 2012*). For instance, in certain contexts, a reactive behavioural type is associated with slower, less exploratory behaviour and less persistence, while a proactive behavioural type is associated with faster exploratory behaviour and more persistence (*Sih & Del Giudice, 2012*). Performing multiple problem-solving experiments over time can help understand consistency in animals' performance and hence, the effect behavioural types have on the correlates of problem-solving success.

Dogs and their closest living relatives, wolves (*Frantz et al., 2016*), differ strongly in their problem-solving success in various paradigms (*Frank & Frank, 1982*; *Frank et al., 1989*; *Miklósi et al., 2003*; *Udell, Dorey & Wynne, 2008*; *Hiestand, 2011*; *Range & Virányi, 2014*; *Marshall-Pescini, Virányi & Range, 2015*; *Udell, 2015*; *Heberlein et al., 2016*; *Rao et al., 2017*; *Brubaker et al., 2017*; *Marshall-Pescini et al., 2017a*, *2017b*). For instance, wolves were more task-focused, showed more motor diversity, were more persistent and were able to generalize better than dogs in a string-pulling task (*Hiestand, 2011*). They were faster and more successful at obtaining food from puzzle

boxes (*Frank & Frank, 1982*; *Udell, 2015*; *Rao et al., 2017*; *Brubaker et al., 2017*) and performed better at a visual discrimination task than dogs (*Frank et al., 1989*). These differences have partly been attributed to the different ecological niches they live in (*Virányi et al., 2008*; *Range & Virányi, 2013*, *2014*; *Marshall-Pescini, Virányi & Range, 2015*; *Werhahn et al., 2016*; *Marshall-Pescini et al., 2017a*, *2017c*; *Brubaker et al., 2017*). Unlike wolves, dogs live in a human dominated niche (*Marshall-Pescini et al., 2017a*). They may hence rely on humans more than wolves do, both, in terms of social support (*Gácsi et al., 2005*), and possibly as 'problem-solvers.' Authors often describe dogs displaying copious amounts of human directed behaviours during problem-solving experiments. There is ample evidence that when confronted with a problem in the presence of a human, dogs are more likely than wolves to look towards and/or interact with the human instead of engaging in the task (*Miklósi et al., 2003*; *Udell, 2015*; *Brubaker et al., 2017*).

Two hypotheses might explain why dogs engage and persist less than wolves in these situations. First, it is possible that prior experience with humans, who often solve problems for dogs, drives the dogs' behaviour. In the human-dominated niche that dogs live in, humans often provide support in all important domains including access to resources such as food (*Marshall-Pescini et al., 2017a*). Hence, dogs might expect humans to solve problems for them and thus turn to humans for help without trying very hard to solve problems by themselves. However, differences in problem-solving success are visible even in dogs and wolves that have identical experience with humans (*Gácsi et al., 2009*; *Virányi & Range, 2011*; *Range & Virányi, 2014*; *Marshall-Pescini, Virányi & Range, 2015*; *Marshall-Pescini et al., 2016*, *2017c*; *Heberlein et al., 2016*; *Rao et al., 2017*).

The second, likelier hypothesis that may explain differences in dogs' and wolves' problem-solving performance, is that adaptations to their respective feeding ecologies (*Fleming et al., 2017*) have resulted in the two species evolving differences in their correlates of problem-solving success, particularly in persistence. Wolves are primarily hunters (*Fleming et al., 2017*) with low success rates (between 10% and 49%) and need to be highly persistent to survive (*Mech, Smith & MacNulty, 2015*). Dogs, however, are primarily scavengers (*Marshall-Pescini et al., 2017a*; *Fleming et al., 2017*), dependant mostly on human refuse (*Atickem, Bekele & Williams, 2009*; *Vanak & Gomper, 2009*; *Newsome et al., 2014*; *Marshall-Pescini et al., 2017a*; *Fleming et al., 2017*) and may not need to be as persistent. Accordingly, in a problem-solving experiment with a human present, dogs might be less persistent, give up earlier than wolves, and then, as there is nothing else to do, explore the test environment, do nothing, or turn towards the human. Following this reasoning, turning to humans might not be a strategic choice to obtain help or support instead of solving the task independently, as has been previously suggested (*Miklósi et al., 2003*; *Gácsi et al., 2005*; *Persson et al., 2015*; *Konno et al., 2016*), but rather a consequence of reduced persistence (*Rao et al., 2017*). Overall, while the socioecology-based hypothesis postulates fundamental differences in motivation (regardless of human presence), the human-reliance hypothesis suggests that, while dogs and wolves might have similar problem-solving skills when alone, dogs turn towards humans as an alternative problem-solving strategy instead of solving problems by themselves.

A first step towards disentangling these hypotheses and better quantifying persistence without direct human influence on dogs' and wolves' performance is to conduct problem-solving tasks in the absence of humans with dogs and wolves whose rearing history and human exposure is controlled for. *Udell (2015)* headed in this direction by testing subjects in three conditions—alone, with a silent human and with an encouraging human. Wolves were more persistent than pet dogs in the task even when alone, which suggests that dogs' may have a 'generalized dependence on humans' (p. 1). However, the authors highlighted that such dependence may have been a result of differences in the life experiences that the pet dogs and hand-reared wolves had. Pet dogs may have been discouraged by their owners to 'problem-solve' the trash-can or kitchen drawers, which may have resulted in dogs being inhibited when confronting a novel object. Differences in life experience are in fact known to affect problem-solving in dogs: highly trained dogs (agility, retriever, search and rescue) showed more independent problem-solving abilities than untrained pet dogs, who, conversely, looked towards their owners longer in such tasks (*Marshall-Pescini et al., 2008*).

Here, we presented similarly raised and kept pack-living dogs and wolves with two different unsolvable tasks in the absence of humans on two separate occasions. Each task consisted of an object baited with food inaccessible to the animal. To avoid animals' expectations regarding the role of a human in the task, we presented the object in their home enclosure, where humans rarely enter. Humans entering the home enclosure is associated with a routine enrichment procedure where the animals are shifted out of the enclosures, humans scatter food in the enclosures, leave, and then shift the animals back in. Apart from removing the expectation of human presence, using an enclosure associated with the enrichment procedure (which is familiar to all animals) guaranteed a similar motivational state for all subjects. Furthermore, because food motivation is known to influence problem-solving behaviour (*Laland & Reader, 1999*; *Sol, Griffin & Bartomeus, 2012*; *Griffin, Diquelou & Perea, 2014*; *Griffin & Guez, 2014*), we tested subjects early in the morning without feeding them the evening prior to the test. Finally, as food motivation is influenced by food quality (*Fontenot et al., 2007*; *Dufour et al., 2012*; *Hillemann et al., 2014*); we used high value food (based on a previously performed preference test) for testing (*Rao et al., 2018*).

We measured persistence as the time spent manipulating the presented objects. We predicted that if human presence during testing and/or general differences in dog–wolf experiences with humans (*Udell, 2015*) are the main factors responsible for wolves' greater persistence in problem-solving experiments, dogs and wolves would not differ significantly in their persistence in the current study. If, however, adaptations to the respective feeding niches play a bigger role than their experience with humans, wolves would be significantly more persistent than dogs.

Although several studies have compared species (*Griffin & Guez, 2014*) and evaluated the effect of different environments on problem-solving behaviours, fewer studies have also examined how problem-solving correlates are interconnected (birds: (*Griffin & Guez, 2014*), mammals: (*Thornton & Samson, 2012*; *Benson-Amram & Holekamp, 2012*; *Borrego & Gaines, 2016*)). Therefore, in the current study, apart from persistence, we also measured motor diversity (the number of different object-directed manipulative

behaviours exhibited) when subjects attempted to extract the food from the presented objects, the latency for subjects to contact each object (contact latency; typically used as a measure of neophobia (*Griffin & Guez, 2014*)) and the body posture (low-insecure vs. high-confident) subjects exhibited during approach and manipulation.

Studies have found that animals that spend longer engaged in a task also tend to utilize a greater variety of behaviours (*Griffin, Diquelou & Perea, 2014*; *Logan, 2016*; *Johnson-Ulrich, Johnson-Ulrich & Holekamp, 2018*). In line with this, we expected to find a positive correlation between persistence and motor diversity. The relationship between persistence and contact latency may be more multifaceted, as contact latency could be a measure of neophobia or a measure of (dis)interest in an object. To try disentangling these possibilities, we included body postures when analysing the contact latency data. If contact latency was a measure of neophobia, we expected it to be higher in subjects that showed an insecure body posture (known to be related to fear (*Marshall-Pescini et al., 2017c*)) during approach. If no such relationship emerged, it may be that contact latency was a measure of the animal's interest in the task.

*Sih & Del Giudice (2012)* proposed that persistence, neophobia and interest may form parts of a behavioural syndrome. If these are indeed personality traits, they would be correlated with each other and be stable over time and context (*Réale et al., 2007*). Hence, independently of whether contact latency is a measure of neophobia or interest, we expected it to be negatively correlated with persistence in both species. Finally, we evaluated whether individual consistency in persistence and in contact latency would emerge across the two tasks. Considering studies suggesting that these may indeed be personality traits (*Sih & Del Giudice, 2012*; *Johnson-Ulrich, Johnson-Ulrich & Holekamp, 2018*), we predicted that our subjects would indeed be consistent in their persistence and contact latency across tasks.

To sum up, our study had three aims: (1) to test hypotheses about why dogs and wolves (with controlled rearing history and human exposure) differ in their persistence, (2) to assess relationships between the correlates of problem-solving success and (3) test subjects' consistency in performance across tasks.

## MATERIALS AND METHODS

### Ethics statement

Special permission to use animals (wolves) in such cognitive studies is not required in Austria (Tierversuchsgesetz 2012). The 'Tierversuchskommission am Bundesministerium für Wissenschaft und Forschung (Austria)' allows research without special permissions regarding animals. We obtained ethical approval for this study from the 'Ethik und Tierschutzcommission' of the University of Veterinary Medicine (Protocol number ETK-07/08/2016).

### Subjects

We tested 17 adult mixed-breed dogs (7 F, 10 M; mean age + SD = 4 + 1.6 years) (*Canis lupus familiaris*) and 12 adult Grey wolves (4 F, 8 M; mean age + SD = 6.3 + 1.7 years) (*Canis lupus*) raised and kept similarly in conspecific packs at the Wolf Science Centre, Austria

**Table 1** Subjects.

| Subject | Species | Sex | Date of birth | Age when tested |
|---------|---------|-----|---------------|-----------------|
| Amarok | Wolf | M | 04/04/2012 | 4.7 |
| Aragorn | Wolf | M | 04/05/2008 | 8.3 |
| Chitto | Wolf | M | 04/04/2012 | 4.3 |
| Geronimo | Wolf | M | 02/05/2009 | 7.3 |
| Kaspar | Wolf | M | 04/05/2008 | 8.6 |
| Kenai | Wolf | M | 01/04/2010 | 6.6 |
| Nanuk | Wolf | M | 28/04/2009 | 7.3 |
| Shima | Wolf | F | 04/05/2008 | 8.4 |
| Tala | Wolf | F | 04/04/2012 | 4.3 |
| Una | Wolf | F | 07/04/2012 | 4.3 |
| Wamblee | Wolf | M | 18/04/2012 | 4.5 |
| Yukon | Wolf | F | 02/05/2009 | 7.3 |
| Asali | Dog | M | 15/09/2010 | 5.9 |
| Banzai | Dog | M | 02/04/2014 | 2.4 |
| Binti | Dog | F | 15/09/2010 | 5.9 |
| Bora | Dog | F | 02/08/2011 | 5.0 |
| Enzi | Dog | M | 02/04/2014 | 2.3 |
| Gombo | Dog | M | 21/03/2014 | 2.4 |
| Hiari | Dog | M | 21/03/2014 | 2.4 |
| Imara | Dog | F | 21/03/2014 | 2.4 |
| Layla | Dog | F | 03/08/2011 | 5.1 |
| Maisha | Dog | M | 18/12/2009 | 6.6 |
| Meru | Dog | M | 01/10/2010 | 5.8 |
| Nia | Dog | F | 22/07/2011 | 5.0 |
| Nuru | Dog | M | 24/06/2011 | 4.9 |
| Panya | Dog | F | 02/04/2014 | 2.4 |
| Pepeo | Dog | M | 02/04/2014 | 2.3 |
| Sahibu | Dog | M | 21/03/2014 | 2.4 |
| Zuri | Dog | F | 24/06/2011 | 5.1 |

(henceforth referred to as the 'WSC') (Table 1). The experiment lasted from October 2016 to February 2017. Subjects were hand-raised with conspecifics in peer groups by humans (dogs were raised separately from wolves and at different times). The animals had continuous access to humans who bottle-fed and later hand-fed them in the first 5 months of their life. During the first weeks of puppyhood, the animals were kept inside. They had free access to a 1,000 m$^2$ outdoor, 'puppy' enclosure from their second month on, and were moved to 2,000–8,000 m$^2$ 'living' enclosures at 5 months of age. The animals, as adults, live in these larger 'home enclosures'. Packs are regularly moved from one home enclosure to another for logistic reasons (such as to make it easier to walk an animal on leash from its home enclosure to a test conducted indoors, or to a touristic event). All packs have resided in all home enclosures.

Every enclosure is equipped with bushes, trees, logs, shelters and permanent drinking water installations. While humans are not continuously present in living enclosures,

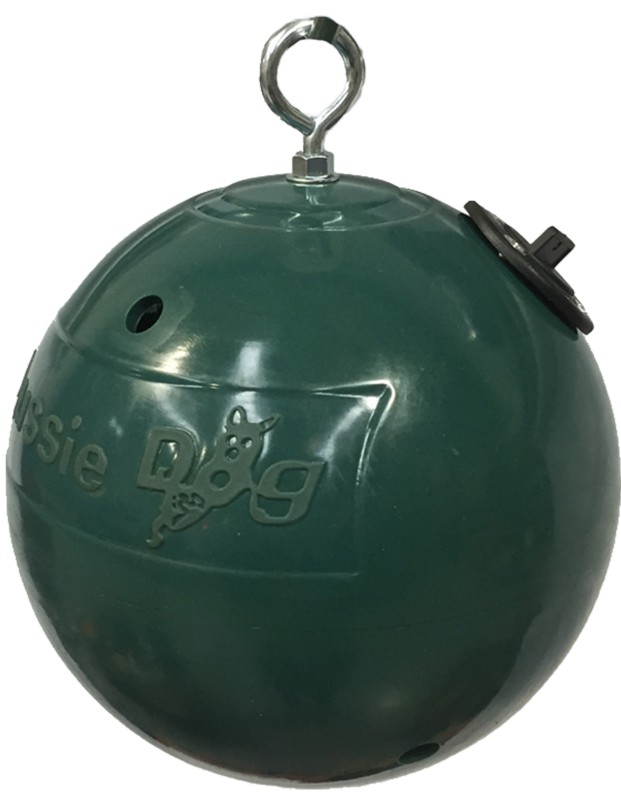

**Figure 1  Commercially available lion feeder ball.** Image credit: Aussie Dog Products (https://aussiedog. com.au/product/lion-feeder-ball).               

all animals have regular social contact with humans as all animals voluntarily participate in cognitive and behavioural experiments, and/or training, and/or other social events at least once a day. Animals are rewarded with food for participating in these activities. This routine ensures that they are cooperative and attentive towards humans and allows weekly veterinary checks without sedation. All animals at the WSC are intact and males are vasectomized. Over the course of their lives, all animals at the WSC have participated in the same behavioural and cognitive experiments and have participated in the same training activities.

## Apparatus

One object (henceforth referred to as the 'ball') was a perforated, hard plastic sphere 24 cm in diameter, weighing 1.5 kg (commercially available 'Lion Feeder Ball' from www.ottoenvironmental.com) (Fig. 1). The other object was a modified, perforated PVC sewage pipe (22 cm in diameter, 40 cm in length henceforth referred to as the 'pipe') (Fig. 2). Prior to the test, each object was baited with large chunks of strongly smelling sausage and meat out of sight of the subject.

## Experimental setup

Before a test session began, we anchored one of the objects using a 30 cm long metal chain to a camping peg driven into the ground in the subjects' home enclosure and marked a two m radius around it with a commercially available, bright red timber marking spray.

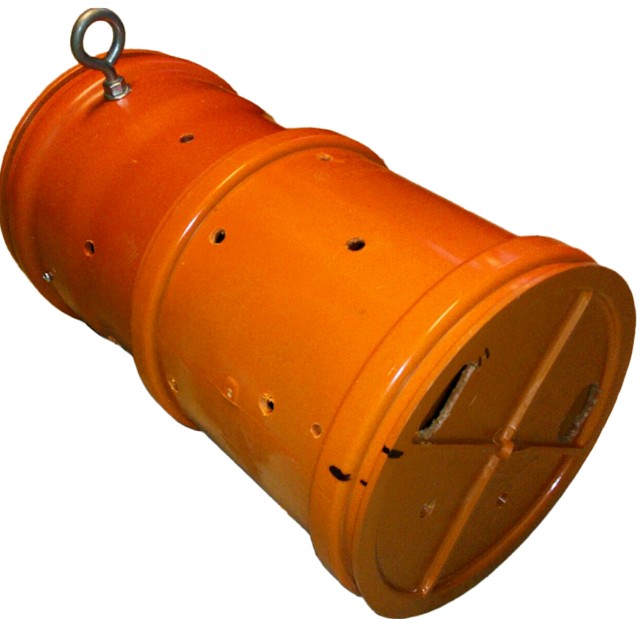

**Figure 2 Modified sewage pipe.** Photo credit: Akshay Rao.

This was done out of sight of the test subject. The peg was positioned such that any interactions the subject had with the object could be recorded from multiple angles without any visual obstructions. Two video cameras (recording at 1,920 × 1,080 pixels at 50 progressive frames per second) and one smartphone (Samsung Galaxy Note 2) were mounted on tripods outside the enclosures at three different angles. We used 'IP Webcam,' a freely available app developed by Pavel Khlebovich (http://ip-webcam.appspot.com/) to remotely monitor the trial, whilst staying out of sight of the subject during the entire procedure.

Subjects were tested in their home enclosure as they least expect a human to be present inside. Tests at the WSC are normally conducted in specific 'testing enclosures' and humans only visit the animals in the home enclosures in very specific contexts (i.e. pack visits, animal care, and short, training demonstrations during public guided tours for visitors and tourists). Subjects were in different home enclosures when they were tested with each object.

## Procedure

We tested subjects individually between 7:00 and 10:00 a.m. One animal per pack was tested per session, and two to three sessions were conducted per week (never on consecutive days). To ensure high food motivation, the subjects were not fed the evening prior to testing. Before the test, we shifted the entire pack out of their home enclosure into an empty enclosure from where their home enclosure was out of sight. The test object was placed in the subjects' home enclosure (see 'Experimental Setup') after which, the focal subject was led back into the home enclosure. We started the test session when the animal entered the two m-radius (see 'Start' in Table 2).
**Table 2 Definitions of coded behaviours.**

| Behaviour | Definition |
|---|---|
| **Approach posture** | |
| Neutral | Body relaxed, tail relaxed below the plane of the back. |
| Confident | Body rigid or relaxed, tail above or at the same level of the plane of the back. |
| Insecure | Tail between the legs (and wagging), and/or back (slightly) lowered, ears can be rearward, and the head can be lowered, approach can be jerky and/or cautious. |
| Friendly | Body relaxed, tail wagging horizontal or below the plane of the back. |
| **Manipulation posture** | |
| Insecure | Tail between the legs, even wagging, or back lowered, ears can be rearward, and the head can be lowered, body can be rigid, and movement can be jerky. |
| Friendly | Tail wagging, not between the legs. |
| Confident | Body rigid or relaxed, tail above or at the same level of the plane of the back. |
| **Behavioural states** | |
| Sniff | The subject smells or attempts to smell the object with its snout less than 10 cm from the object. |
| Manipulating | The subject physically manipulates the object using its paws, snout, mouth or any combination of the three and shows any of the 'Manipulative Behaviours.' |
| **Markers** | |
| Start | The subject places a paw inside the marked two m radius. |
| End | 1. The subject stops manipulating the object for 5 min or |
| | 2. The subject has not started manipulating the object for 5 min after making 'First Contact' or |
| | 3. The subject has not made 'First Contact' 5 min after 'Start.' |
| **Manipulative behaviours** | |
| Nose | The subject moves the apparatus or tries to lift it with only its nose. |
| Bite | The subject bites the object/raises the object off the ground by holding it with its mouth by the chain, by the object's surface or edges, or by the screws/pulls either the chain, the screws or the object's surface or edges with its mouth. |
| 1 Paw | The subject places its paw on the object without scratching it/uses one paw to scratch at the top of the object while attempting to move the object towards itself/away from itself/laterally. |
| 1 Paw & bite | The subject places its paw on the object and simultaneously bites the object. |
| 1 Paw & nose | The subject sniffs/lifts/pushes the object with its nose or licks the object while also manipulating the object with one paw. |
| Paws on | The subject places both paws on the top of the object and presses the object down. |
| Scratch | The subject scratches the object's surface with both its paws by alternating them (without its paws touching the ground). |
| Scratch & bite | The subject scratches at the object with both its paws while simultaneously biting it. |
| Hold & bite | The subject holds and stabilizes the object with both paws on the sides of it or on the top of it for the pipe, while biting it on top. |

(Continued)

| Table 2 (continued). | |
|---|---|
| **Behaviour** | **Definition** |
| Dig | The subject uses one or both of its paws to dig at the ground in immediate proximity of the object. |
| **Other behaviours** | |
| Pee | The subject urinates on the object or on or inside the circle. |
| Lick | The subject licks the object. |
| Bark | The subject vocalizes at the object. |
| Withdraw | The subject jumps away from the object in a neutral or insecure posture after looking at it, approaching it, sniffing it, or manipulating it. |
| Lay down | The subject lays down or sits next to the object or inside the marked radius. |

The subject was given five minutes to make first contact with the object. We defined 'First Contact' as the first time the subject touched or sniffed the object (in case of a sniff, when the nose was within five cm of the object). In case there was no 'First Contact' within 5 min, the test session was terminated. If the subject did not interact (i.e. 'Sniff' or 'Manipulate' the object—see Table 2 for definitions of all behaviours and behavioural states) with the apparatus at all for 5 min after 'First Contact,' the session was terminated. After the subject started interacting with the object, there was no limit on the duration it could continue to do so. Each time the subject stopped interacting with the object, we started a 5 min countdown. If the subject resumed interacting with the object before the countdown expired, we let the test session continue and reset the 5 min countdown timer. If the subject did not resume interacting with the object by the time the countdown expired, we terminated the test session. To simplify, if a subject started interacting with the object, it could continue doing so for an infinite duration and pause as many times as it liked, as long as each pause was shorter than 5 min; once it paused for more than 5 min, the test session ended.

After the session ended, we shifted the subject out of the home enclosure and retrieved the object. We carefully washed each object after each session to remove any possible odour cues left by the previously tested subject. Each subject was tested first with the ball and then, one and a half to three months later, re-tested with the pipe. Two wolves, Chitto and Tala, had to be tested with the pipe 6 months after their test with the ball due to the onset of the mating season. As we needed to keep our study comparable to a complementary study with free-ranging and pet dogs which were presented with only the ball (M. Lazzaroni et al., 2018, unpublished data), we were unable to counterbalance the presentation order of the two objects. We used each object only once per subject to avoid object-specific learning effects (e.g. to avoid subjects learning 'food from inside this particular green sphere cannot be extracted').

## Behavioural coding

We recorded all tests on video and coded behaviours using Solomon Coder beta 100926 (a behaviour coding software developed by András Péter, Dept. of Ethology, Budapest; www.solomoncoder.com). We categorized manipulative behaviours based on the number of body parts involved and the nature of the behaviour. For instance, we differentiated

between using paws to hold an object and to scratch vigorously at the object while the subject simultaneously gnawed at it. 'Holding' an object with the paws may have added stability which probably made 'Biting' more efficient, while 'Scratching' may not have added stability, but was perhaps a different strategy to extract the food within the object. The coded behaviours and their definitions are summarized in Table 2. See the Supplementary Video for an example of each behaviour. We defined 'Persistence' as the time (in seconds) a subject spent in the 'Manipulating' behavioural state. We defined 'Contact Latency' as the time (in seconds) a subject took from 'Start' to 'First Contact.' We defined 'Motor Diversity' as the number of unique 'Manipulative Behaviours' shown by a subject.

## Analyses

We excluded one dog (Gombo) from the analyses for the pipe as he extracted some food from the object due to an apparatus malfunction (a piece of meat that we used had several long fibres that were too close to the holes in the apparatus, which allowed Gombo to easily grab them and pull a piece of meat out through one of the holes). We excluded one wolf (Una) from the latency analyses for the ball as her contact latency was an outlier (28 s; $G = 5.09$, $U = 0.007$, $P < 0.001$) (potentially because she was tested at the onset of the mating season). We excluded one dog (Nuru) from the analyses of the pipe as he was very unusually overly persistent with the pipe, making his manipulation duration an outlier (1,361 s, $G = 3.10$, $U = 0.63$, $P = 0.008$). We used Grubbs tests (*Grubbs, 1950*) with the package 'outliers' (version 0.14) (*Komsta, 2006*) in R version 3.4.3 (*R Core Team, 2017*) to confirm that these individuals were indeed outliers. See the Supplementary Material for how results changed when these latter two individuals were included in the analyses. All other subjects were included in the analyses (Ball: $N = 11$ wolves, 17 dogs, Pipe: $N = 12$ wolves, 15 dogs).

We used inter-class correlations (*Shrout & Fleiss, 1979*) implemented with the 'psych' package (version 1.7.8) (*Revelle, 2017*) in R version 3.4.3 (*R Core Team, 2017*) to calculate inter-observer reliability. A second coder coded 20% of the data and all variables achieved reliability coefficients between 0.89 and 0.99 between the two coders.

We first used an exploratory, principal component analysis (PCA) for each object to understand our data. Performing several univariate analyses may not have allowed us to understand the combined effect of all explanatory variables on our subjects' task performance. As we were primarily interested in variables that have previously been shown to relate to problem-solving success, we included persistence, motor diversity, latency to contact, approach posture and likelihood of manipulation as explanatory variables. While we could have included several more variables (such as the frequencies of each manipulative behaviour), we chose to restrict the number of explanatory variables due to our relatively small dataset. We used the PCAmixdata package (version 3.1) (*Chavent et al., 2014*) in R (version 3.5.1) (*R Core Team, 2018*) which is designed for analysing multivariate data that is a mixture of continuous, discrete and categorical variables.

The PCAmixdata analysis algorithm classified subjects based on our explanatory variables which did not include 'Species.' The rationale behind leaving species out of

the analysis was to allow the algorithm to classify subjects based purely on task performance and without any pre-existing bias. This way, if, for example there were distinct behavioural differences between the two species, it would result in data-point clusters composed entirely of dogs and entirely of wolves, and each cluster would have different values of one or more behavioural variables. Conversely, if there were no differences, it may still result in clusters with different variable values, but these clusters would be mixtures of dogs and wolves. We ran a separate multivariate analysis for each object as including data from both objects in one analysis made it difficult to meaningfully interpret the clusters' structures. Separating the two objects allowed us to analyse whether subjects performed similarly with both objects. Additionally, we applied an orthogonal rotation procedure to each PCA to make interpretation easier. We used the 'PCArot' function, which uses a generalization of the varimax procedure for mixed data (*Chavent, Kuentz-Simonet & Saracco, 2012*). This procedure helps associate variables with a selected number of principal components (or dimensions) more clearly by providing either large (almost 1) or small (almost 0) loadings. While the variable loadings on each dimension (and hence the variance explained by each dimension) change after rotation, the total variance explained by the selected dimensions remains unchanged.

The PCA gave us useful insights into patterns in our data, but did not let us test whether there was a statistically significant difference in dogs' and wolves' performance when interacting with the two objects (we did not make any inferences based on the PCA results). Hence, we further analysed persistence, motor diversity and contact latency individually using generalized additive models for location, scale and shape ('gamlss' version 5.1-0) (*Stasinopoulos & Rigby, 2007*) in R version 3.5.1. We used the 'gamlss.Dist' package (version 5.0-6) to fit distributions to our data. We evaluated the distribution of each response variable and specified the best fitting distribution in the models. We evaluated model fits both by their generalized Akaike information criteria (AIC; *Akaike, 1974*) and by the distribution of the model residual quantile–quantile plots. This approach enabled us to analyse the data without major transformations (data transformations might have negatively affected our interpretations of the results (*Feng et al., 2014*; *Lo & Andrews, 2015*)).

To reduce the risk of our choice of distributions resulting in overfitting models to our data, we validated our models' results by fitting identical models with other probable distributions, and compared models with different distributions but similar AIC values. Further, when our data fit multi-parametric variations of the same distribution equally well, we used the distribution with fewer parameters (e.g. 'Persistence' fit Weibull-1, Weibull-2 and Weibull-3 but we used Weibull-1, as this distribution is described with one parameter as against two or three). Results did not change between models, implying that they were robust against choice of distribution. For the sake of brevity, we have only reported results from models with the best fitting distributions here. Please see the Supplementary Material for the complete distribution selection, model selection and validation processes, outputs, and R scripts. To account for repeated measures, we included the individual as a random effect in all models that included 'Object' as a fixed effect. As our subjects ages varied, we included 'Age' as a fixed effect in all our models. This allowed us to account for any effects age may have on subjects' task performance (*Siwak, 2001*).

When interactions were not statistically significant, we ran a reduced model that included the same fixed and random effects but not the interaction term. We have reported the results from these, reduced models whenever interactions were not significant.

Based on the PCA's results, we used a Fisher's Exact Test in R version 3.5.1 to investigate whether dogs and wolves differed statistically in their likelihood to manipulate the objects. Our PCA suggested that wolves and dogs may differ in their persistence, but that this difference may be influenced by object type. To investigate this, we used a GAMLSS model to evaluate the effects of species, object type and a two-way interaction between them on the response variable persistence. To ensure model convergence, we added a miniscule constant (0.00001) to all persistence values. We fit this model with the Gamma distribution and validated it with the Box-Cox T Original, Weibull and Log-normal distribution. This process allowed us to achieve our first aim of testing our hypothesis about dog–wolf differences. We left motor diversity out of this analysis for two reasons: (1) our hypothesis pertained specifically to differences in *persistence* between dogs and wolves and (2) from our PCA (and from further analysis for our second aim), persistence and motor diversity appeared to be correlated; this collinearity may have negatively impacted our interpretation of model results (*Graham, 2003*).

For our second aim, we focussed on understanding the relationships between the correlates of problem-solving success within dogs and within wolves. We analysed data for both species separately by running separate GAMLSS models for dogs and wolves. The rationale behind this decision was that the only hypothesis we had pertaining to dog–wolf differences was about persistence and did not encompass other behavioural measures.

We ran two GAMLSS models with contact latency as the response variable. Our PCA suggested that contact latency may be related to object type, and that approach posture and persistence may influence contact latency differently in both objects. Hence, for dogs, we included object type, persistence, approach posture and two two-way interactions (object type by persistence and object type by approach posture) as explanatory variables. For dogs, we fit the model with the Inverse Gaussian distribution and validated it with the Inverse Gamma, Log-normal and Gamma distribution. For wolves, we fit the model with the Log-normal distribution and validated it with the Gamma, Weibull and Box-Cox Cole-Green distribution.

We ran two GAMLSS models with motor diversity as response variable. As the PCA suggested that persistence and motor diversity may be correlated, and because this correlation appeared slightly different between the two objects, we included persistence, object type and a two-way interaction between persistence and object type as explanatory variables. For dogs, we fit the model with the Zero Adjusted Poisson distribution and validated it with the Zero Inflated Poisson, Zero Adjusted Negative Binomial (Type I) and Zero Inflated Negative Binomial (Type I) distribution. For wolves, we fit the model with the Poisson distribution and validated it with the Zero Adjusted Poisson, Negative Binomial type I and Generalized Poisson Distribution.

Our last aim was to test subjects' consistency in performance between the two tasks. As we had not restricted the duration a subject could manipulate both objects, and as contact latency could have varied due to the layout of the enclosure subjects were tested in,

**Table 3 Summary of the PCA results for the ball.**

**Before orthogonal rotation**

| Dimension | Eigenvalue | Variance explained | | Variable loadings | | | | |
|---|---|---|---|---|---|---|---|---|
| | | Individual | Cumulative | Contact latency | Persistence | Motor diversity | Approach posture | Manipulation likelihood |
| 1 | 2.1059 | 42.1187 | | 0.1925 | 0.6168 | 0.7670 | 0.0353 | 0.4944 |
| 2 | 1.0595 | 21.1904 | 63.3091 | 0.5968 | 0.1596 | 0.0696 | 0.0934 | 0.1401 |
| 3 | 0.9985 | 19.9693 | 83.2783 | 0.0046 | 0.0454 | 0.0496 | 0.8473 | 0.0516 |
| 4 | 0.6380 | 12.7605 | 96.0388 | 0.2010 | 0.1117 | 0.0156 | 0.0215 | 0.2882 |
| 5 | 0.1981 | 3.9612 | 100.0000 | 0.0051 | 0.0665 | 0.0982 | 0.0025 | 0.0257 |
| **After orthogonal rotation** | | | | | | | | |
| 1 | 1.8086 | 36.1719 | | 0.0001 | 0.8191 | 0.8548 | 0.0008 | 0.1337 |
| 2 | 1.3214 | 26.4285 | 62.6003 | 0.7849 | 0.0003 | 0.0312 | 0.0000 | 0.5050 |
| 3 | 1.0339 | 20.6780 | 83.2783 | 0.0089 | 0.0023 | 0.0002 | 0.9751 | 0.0474 |

absolute persistence and latency values may not have been meaningfully comparable. Hence, we scaled these values from 0 to 1 in each task separately using the following formula for both variables: $V_s = \frac{V_i - \text{Min}(V_{\text{all}})}{\text{Max}(V_{\text{all}}) - \text{Min}(V_{\text{all}})}$ where $V_s$ = scaled value (persistence or contact latency), $V_i$ = individual's unscaled value, Min/Max $(V_{\text{all}})$ = the minimum/maximum values for that object. We used a Spearman's rank correlation on the scaled persistence and scaled contact latency data to test whether subjects were consistent in their persistence and contact latency between the two objects. We calculated a consistency score for persistence and contact latency by taking the absolute value of the difference between subjects' scaled persistence scores (or scaled contract latency scores) for the ball and for the pipe. We used separate GAMLSS models to assess the effect of species on the consistency scores for persistence and contact latency. For persistence, we fit the model with the Generalized Beta Type 1 distribution and validated it with the Logit Normal distribution. For contact latency, we fit the model with the Simplex distribution and validated it with the Logit Normal and Beta Original distributions.

## RESULTS

### Multivariate analyses

The PCA for the ball produced five dimensions, the first three of which explained 83.28% of the variance in our data. Pre and post orthogonal rotation results are summarized in Table 3. The rotation significantly improved variable loadings on dimensions 1 and 3. Hence, we investigated these dimensions further. We found that dogs and wolves segregated into two near-distinct clusters along dimension 1, but not along dimension 3 (Fig. 3A). Persistence (0.82) and motor diversity (0.85) loaded very strongly on dimension 1 (Fig. 3B), suggesting that the segregation between dogs and wolves was likely due to differences in either persistence, motor diversity, or both, and that these two variables may be correlated. We found two distinct clusters along dimension 3, but each of these clusters were composed of both dogs and wolves. Approach posture loaded very

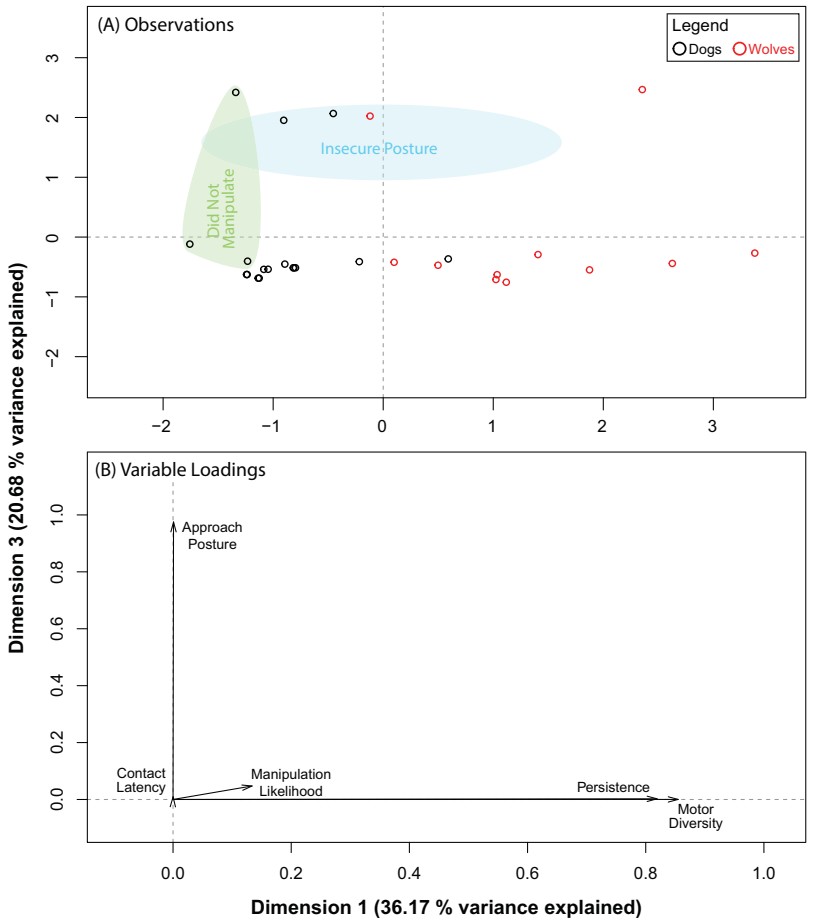

**Figure 3 Results for the PCA for the ball.** (A) shows where each data point placed with respect to dimensions 1 and 3 (after orthogonal rotation). (B) shows how behavioural variables loaded on dimensions 1 and 3 (after orthogonal rotation).

strongly (0.98) on dimension 3 (as did contact latency, but to an almost negligible extent: 0.009). This suggested that there may be a very weak (if any) connection between contact latency and approach posture, and that neither of these variables were likely to be responsible for dog–wolf differences.

The PCA for the pipe also produced five dimensions, the first three of which explained 87.03% of the variance in our data. Pre and post orthogonal rotation results are summarized in Table 4. We investigated dimensions 1 and 2 further as the rotation significantly improved variable loadings on them. Unlike with the ball, wolves and dogs did not segregate into distinct clusters along either dimension (Fig. 4A). Persistence (0.68) and motor diversity (0.84) loaded strongly on dimension 1 (Fig. 4B), suggesting that these variables may be correlated. Like with the ball, approach posture and contact latency loaded strongly on different dimensions (Table 4; Fig. 4B). This supported results with the ball and suggested that there may not be a connection between contact latency and approach posture, and that neither variable contributed to dog–wolf differences.

**Table 4 Summary of the PCA results for the pipe.**

**Before orthogonal rotation**

| Dimension | Eigenvalue | Variance explained | | Variable loadings | | | | |
|---|---|---|---|---|---|---|---|---|
| | | Individual | Cumulative | Contact latency | Persistence | Motor diversity | Approach posture | Manipulation likelihood |
| 1 | 2.3801 | 47.6022 | | 0.2343 | 0.6770 | 0.8871 | 0.1554 | 0.4262 |
| 2 | 1.3100 | 26.1996 | 73.8018 | 0.4153 | 0.0922 | 0.0064 | 0.4906 | 0.3055 |
| 3 | 0.6613 | 13.2266 | 87.0284 | 0.2499 | 0.0330 | 0.0275 | 0.2972 | 0.0537 |
| 4 | 0.5132 | 10.2645 | 97.2929 | 0.0985 | 0.1593 | 0.0005 | 0.0567 | 0.1983 |
| 5 | 0.1354 | 2.7071 | 100.0000 | 0.0020 | 0.0386 | 0.0785 | 0.0000 | 0.0163 |
| **After orthogonal rotation** | | | | | | | | |
| 1 | 2.0270 | 40.5395 | | 0.0525 | 0.6832 | 0.8414 | 0.0062 | 0.4437 |
| 2 | 1.1986 | 23.9721 | 64.5116 | 0.0002 | 0.0032 | 0.0243 | 0.9370 | 0.2339 |
| 3 | 1.1258 | 22.5167 | 87.0284 | 0.8468 | 0.1158 | 0.0553 | 0.0001 | 0.1079 |

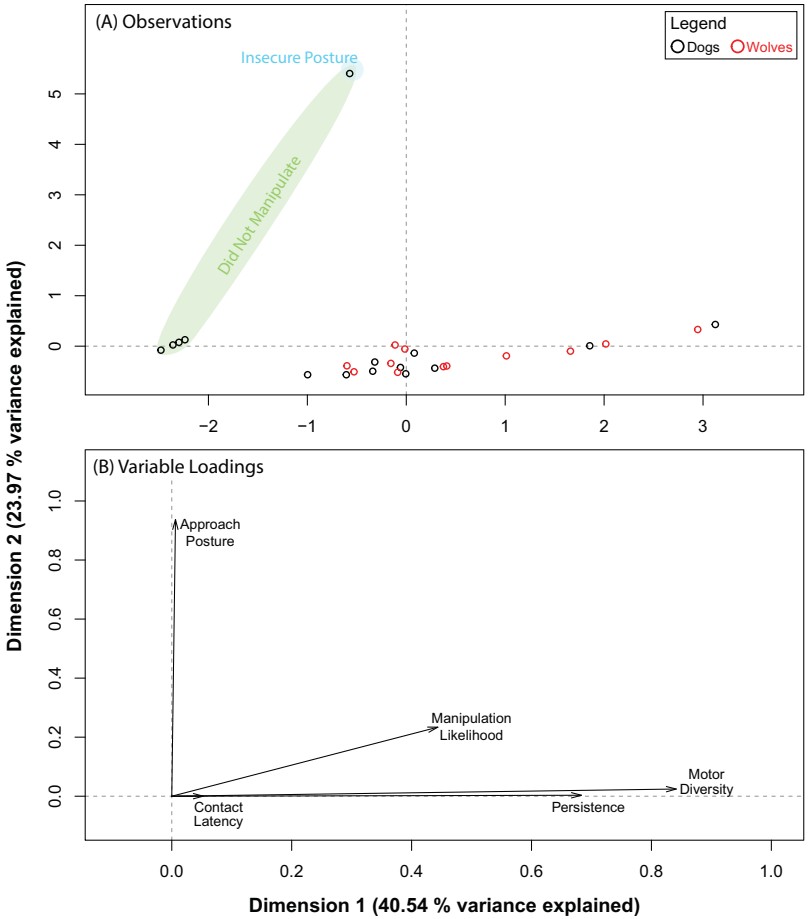

**Figure 4 Results for the PCA for the pipe.** (A) shows where each data point placed with respect to dimensions 1 and 3 (after orthogonal rotation). (B) shows how behavioural variables loaded on dimensions 1 and 3 (after orthogonal rotation).

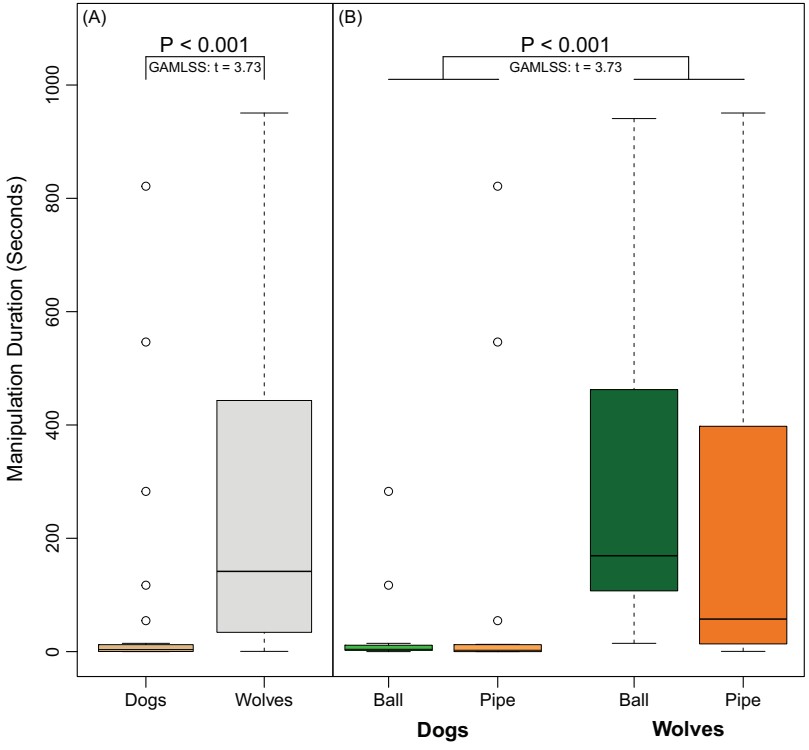

**Figure 5 Differences in persistence between dogs and wolves.** (A) shows the time (in seconds) dogs and wolves spent manipulating both apparatuses combined. (B) shows the time (in seconds) dogs and wolves spent manipulating each object separately. Circles indicate data points that were outside the upper quartile plus 1.5 times the interquartile distance.

## Differences in persistence between wolves and dogs (GAMLSS)

Overall, 14 out of 17 dogs manipulated the ball and 10 out of 15 dogs manipulated the pipe. In contrast, all 11 wolves manipulated the ball and all 12 wolves manipulated the pipe. Wolves were significantly more likely to manipulate objects than dogs (Fisher's Exact Test, Odds Ratio = 0.0, 95% CI [0.00–0.71], $P = 0.015$). Though the PCA suggested that persistence may have been affected by object type, the interaction between species and object was not significant (GAMLSS: $t = -1.47$, $P = 0.15$). Wolves were more persistent than dogs (GAMLSS: $t = 3.73$, $P < 0.001$) in their manipulation of the objects regardless of object-type (Fig. 5A). Neither subjects' age (GAMLSS: $t = 0.76$, $P = 0.45$) nor object type (GAMLSS: $t = 1.06$, $P = 0.29$) affected persistence (Fig. 5B).

## Relationship between correlates of problem-solving within dogs and within wolves

Contact latency decreased with persistence in both, dogs (GAMLSS: $t = -4.35$, $P < 0.001$) and wolves (GAMLSS: $t = -3.42$, $P < 0.01$). Neither the interaction between object type and persistence (GAMLSS; Dogs: $t = 1.91$, $P = 0.07$, Wolves: $t = -0.96$, $P = 0.35$) nor that between object type and approach posture significantly affected contact latency (GAMLSS; Dogs: $t = -1.32$, $P = 0.20$, Wolves: $t = -1.61$, $P = 0.13$). Neither object type (GAMLSS; Dogs: $t = 1.44$, $P = 0.16$, Wolves: $t = -0.96$, $P = 0.35$) nor approach posture (GAMLSS; Dogs: $t = 0.43$, $P = 0.67$, Wolves: $t = -1.72$, $P = 0.10$) significantly
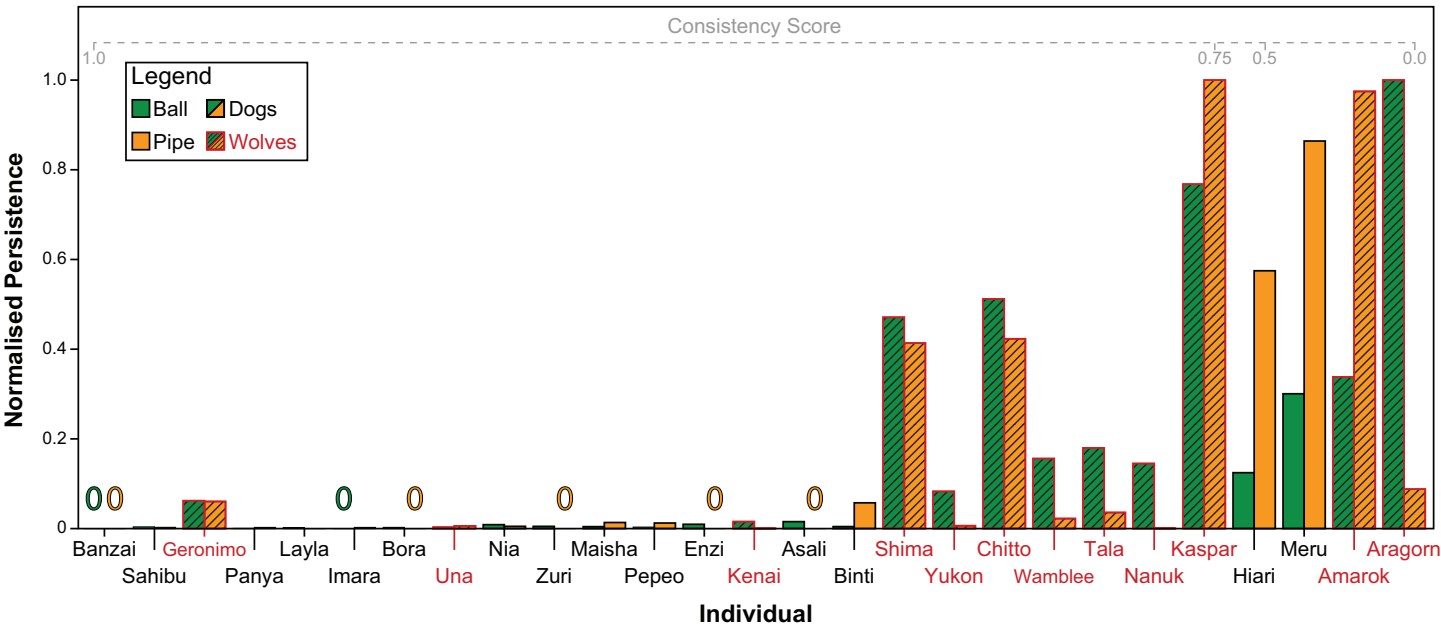

**Figure 6 Every individual's persistence in both tasks, re-scaled from 0 to 1 for comparability.** Green bars indicate persistence with the ball, orange bars indicate persistence with the pipe. Zeros indicate that the individual did not manipulate the object at all. Individuals with red names and hashed bars are wolves, individuals with black names and non-hashed bars are dogs. Individuals are arranged from left to right in descending order of consistency in persistence across tasks.

affected contact latency in either species. Contact latency decreased with age in dogs (GAMLSS: $t = -2.85$, $P < 0.001$) but not in wolves (GAMLSS: $t = -0.04$, $P = 0.97$).

Motor diversity increased with persistence in both, dogs (GAMLSS: $t = 3.74$, $P < 0.001$) and wolves (GAMLSS: $t = 3.72$, $P = 0.001$). The interaction between object type and persistence was not significant (GAMLSS; Dogs: $t = -1.67$, $P = 0.11$, Wolves: $t = 1.62$, $P = 0.12$). Neither object type (GAMLSS; Dogs: $t = -1.74$, $P = 0.09$, Wolves: $t = -1.61$, $P = 0.12$) nor age (GAMLSS; Dogs: $t = -0.58$, $P = 0.57$, Wolves: $t = 1.20$, $P = 0.24$) significantly affected motor diversity in either species.

### Individual consistency

Both subjects' persistence (Spearman's $\rho = 0.71$, $P < 0.001$) and contact latency (Spearman's $\rho = 0.64$, $P < 0.001$) across tasks were significantly correlated. Figure 6 shows individuals' scaled persistence in both tasks. Overall, dogs were significantly more consistent both, in their persistence (GAMLSS: $t = -2.31$, $P = 0.031$) as well as in their contact latency (GAMLSS: $t = -2.62$, $P = 0.02$) than wolves.

For descriptive statistics of both groups' performance in each task and for complete model information, see the Supplementary Material.

### DISCUSSION

We tested similarly raised dogs and wolves with two unsolvable tasks in the absence of humans on two separate occasions. We had three aims: First, to test hypotheses about why dogs and wolves with controlled rearing history and human exposure differ in their persistence in an independent problem-solving task; second, to evaluate relationships

between correlates of problem-solving success in our subjects; and third, to assess our subjects' consistency in task performance. We used two approaches when analysing our data: first, a multivariate PCA, and second, a set of univariate mixed models.

Results from the multivariate approach suggested that wolves were more persistent and had greater motor diversity with the ball than with the pipe. This may have been due to wolves' ability to generalize (*Hiestand, 2011*). Wolves may have learned that trying to solve a task presented in that specific setting was futile and did not persist as long with the pipe, which was presented as the second task. Alternatively, it is possible that a neophobic response may have affected wolves' persistence and motor diversity negatively (*Sol et al., 2011*; *Thornton & Samson, 2012*; *Griffin & Guez, 2014*) with the pipe.

However, wolves' contact latency, persistence and motor diversity did not differ significantly between the ball and pipe when these measures were analysed with mixed models. Accordingly, it is unlikely that a neophobic response affected wolves' persistence and motor diversity. This lack of neophobic response may either be due to the objects themselves not being 'intimidating' enough, or due to our subjects' experience with several novel objects over their lives. It is possible that like in the study by *Moretti et al. (2015)*, contact latency was a measure of interest in novel objects rather than neophobia. While counterbalancing the order in which the two objects were presented would have allowed better control over novelty, neophobia and generalization, we had to ensure that all subjects interacted with the ball first to keep this study comparable to a parallel one being run on free-ranging dogs (where testing an individual repeatedly with a gap of two or more weeks was impossible).

Our first, bottom-up, descriptive, multivariate approach categorized wolves and dogs according to behavioural measures (contact latency, approach posture, manipulation likelihood, motor diversity and persistence). It allowed us to see how our subjects differed in their behaviour and how behavioural aspects may be correlated. Importantly, as 'species' did not factor into this analysis, clusters of dog/wolf data points were exclusively due to behavioural variables. Results from this analysis allowed us to compare dogs and wolves directly in their persistence, and to decide which factors to include when modelling the other behaviour variables. However, as the multivariate analysis was a purely exploratory approach, we made our inferences and conclusions based on mixed models.

When directly comparing dogs' and wolves' persistence in the two tasks using mixed models, our results confirm numerous other studies (*Hiestand, 2011*; *Frank, 2011*; *Udell, 2015*; *Marshall-Pescini et al., 2017a*, *2017b*, *2017c*; *Rao et al., 2017*) that have found wolves to be more persistent than dogs in object manipulation. We found these differences to hold even in the absence of humans during testing, and importantly, with dogs and wolves that have the same level of experience with both, humans, and with interacting with different objects. A potential concern with using food as a motivator in comparative problem-solving studies is that different species may have different preferences for the same food. In our case, wolves and dogs did not differ in their preference for meat and sausage (*Rao et al., 2018*). A related concern is if these tasks truly test persistence, or if they test motivation to work for food. Persistence has been defined as 'task-directed motivation' (*Griffin & Guez, 2014*), but it is important to note that disentangling these
two concepts is virtually impossible (and is not the focus of this study). Overall, our results cannot be explained by dogs (but not wolves) having been inhibited from interacting with objects in their daily lives (e.g. pet dogs), or by dogs preferring to use a social problem-solving strategy in the presence of a human (i.e. by asking for help instead of solving the problem alone), or by differences in dogs' and wolves' preference for the food used to bait the objects.

Contrary to *Siwak (2001)*, we found older dogs to be more interested in test objects. Dogs at the WSC are kept differently from those in the beagle colony at the University of Toronto (which participated in the *Siwak (2001)* study), and potentially also have different life experiences (WSC dogs live in groups while the beagles in Toronto are housed individually). It is possible that at the WSC, older dogs have grown more accustomed to cognitive testing and are more task focussed than younger dogs, who may be more interested in exploring their environment instead.

We suggest that the results (wolves being more persistent than dogs) are in line with the hypothesis that differences in dogs' and wolves' problem-solving performance are due to adaptations to their respective feeding ecologies. Dogs have been proposed to be selected against directly manipulating their environment and potentially for lower persistence (*Hiestand, 2011*), with humans being intermediaries between dogs and their environment (*Frank & Frank, 1985*). Wolves, however, require high levels of persistence to survive in the wild (*David Mech, 1966*; *Mech & Korb, 1978*; *Mech, Smith & MacNulty, 2015*). Further, wolves are more sensitive to their environment (*Hiestand, 2011*); while they are more neophobic, they are also more explorative than dogs (*Moretti et al., 2015*; *Marshall-Pescini et al., 2017c*). Considering that the animals in the current study had the same experience of human provisioning and interaction during object manipulation, we suggest that differences in persistence are more likely due to dogs' and wolves' adaptations to their respective ecological niche.

The current results cannot reveal the extent to which dogs' persistence is affected by their generalist-foraging style and by the active role being played by humans in their feeding ecology (such as humans providing dogs with food (*Sen Majumder et al., 2016*) or actively inhibiting them from interacting with objects, which may be the case with pet dogs). Comparing dog populations with varying levels of experience with humans (such as pet dogs and free-ranging dogs) may help to better understand whether dogs' reduced persistence could be a result of humans inhibiting their interactiveness with objects.

In line with previous studies (*Morand-Ferron et al., 2011*; *Benson-Amram & Holekamp, 2012*; *Huebner & Fichtel, 2015*; *Borrego & Gaines, 2016*), we found motor diversity to be positively linked to persistence in both tasks, in both dogs and wolves. Motor diversity and behavioural flexibility are important during foraging. Being able to employ and switch between different strategies both, when hunting and when scavenging, may increase success rates regardless of foraging style. We found persistence and contact latency to be negatively correlated. Our results are in line with predictions based on the concept of behavioural types (*Sih & Del Giudice, 2012*). Individuals that were faster to contact the apparatus, were presumably more interested and proactive in their approach, and were hence more persistent.

Finally, we found that our subjects were consistent in their persistence and contact latency between the two tasks. Persistence is an important aspect of animal personality (*Gosling, 1998*; *Svartberg, 2002*; *Range, Leitner & Virányi, 2012*; *Sih & Del Giudice, 2012*; *Massen et al., 2013*). We found dogs to be more consistent in their persistence (or lack thereof) and their contact latency than wolves. A likely explanation for this could be that selection against persistence (*Hiestand, 2011*) and direct manipulation of the environment (*Moretti et al., 2015*; *Brubaker et al., 2017*) may have resulted in a more consistent reactive-type personality. Wolves, having faced no such selection, may be more variable in their behaviour. Alternatively, wolves' ability to better generalize and understand that the task is unsolvable may have influenced the consistency in their performance. To disentangle these possibilities, it would be necessary to test subjects in tasks that are similar in concept but in different test settings. Further, utilising multiple tests would provide a better insight into inter-task performance consistency.

Our study was the first to test differences in persistence between similarly raised and experienced dogs and wolves in an unsolvable task in the absence of humans. Past studies have used tasks that have initially been solvable and later become unsolvable. It is possible that persistence may differ between these two designs. The 'unsolvable task' paradigm has been widely used with dogs and wolves (*Miklósi et al., 2003*; *Gácsi et al., 2005*; *Passalacqua et al., 2011*; *Smith & Litchfield, 2013*; *Marshall-Pescini et al., 2013*; *D'Aniello et al., 2015*; *Udell, 2015*; *Rao et al., 2017*). It involves repeatedly allowing a subject to find a solution to a simple foraging task, and then modifying the task to make it unsolvable. Data about persistence are usually collected in the unsolvable trial.
This approach has certain drawbacks when studying the correlates of problem-solving success. First, it reinforces certain manipulative behaviours, potentially reducing the motor diversity that the subject would show in the unsolvable trial. Second, reinforcing task-engagement with solvable trials may potentially increase persistence in the unsolvable trial. A task that is unsolvable from the start may provide a more reliable measure of persistence. Third, as human presence affects dogs' and wolves' behaviour differently during the test, testing subjects in the presence of a human may make directly comparing wolves' and dogs' persistence difficult.

While several studies have investigated problem-solving behaviour in dogs and wolves, few have analysed consistency in problem-solving success in dogs (*Svartberg & Forkman, 2002*; *Svartberg, 2005*), and none have done so in wolves. By testing dogs and wolves in independent problem-solving tasks with and without the presence of a human, using tasks that offer either controlled or random reinforcement, and by using a battery of various physical problem-solving tasks, future studies could improve our understanding of how the domestication process has affected problem-solving behaviour in the two canids, and the role personality traits play in their problem-solving success. Our study provides an interesting starting point in this direction.

## CONCLUSIONS

We compared equally raised and kept pack-living wolves and dogs in an independent problem-solving task using the unsolvable task paradigm in the absence of humans.

Wolves were more likely than dogs to engage in the presented tasks and were more persistent at attempting to extract food from the presented objects. Results from this study support the socioecology-based hypothesis, which suggests that differences in dogs' and wolves' problem-solving performance stem from fundamental differences in the correlates of their problem-solving success, and that these correlates have evolved differently in dogs and wolves due to differences in the two species' feeding ecologies. Further, persistence and motor diversity were positively correlated, and subjects were consistent in their persistence and approach latency across tasks, dogs more so than wolves.

Comparing dog populations that have different experiences with humans (e.g. pets and free-ranging dogs) and testing subjects in identical tasks both, with and without humans present in the test setting may help further disentangle the human-reliance and socioecology-based hypotheses. Using a battery of conceptually similar tests across varying test settings may provide better insight into the role of behavioural types or personality in problem-solving success.

## ACKNOWLEDGEMENTS

The Wolf Science Centre was established by Zsófia Virányi, Kurt Kotrschal and Friederike Range, and we thank all the helpers who made this possible, hence indirectly supporting this research. We thank all animal trainers at the WSC for raising and caring for the animals: Rita Takacs, Marleen Hentrup, Christina Mayer, Marianne Heberlein, Lars Burkart and Cindy Voigt. We thank Giulia Cimarelli and Ashish Sharma for the statistical advice. The authors further thank Royal Canin and many other private sponsors for their support (such as providing food for the animals, donating toys and bedding material for the pups during the raising process, providing equipment for the trainers, etc.) and the Game Park Ernstbrunn for hosting the Wolf Science Centre, providing food for the animals and for personnel support.

### Funding

Sarah Marshall-Pescini and Akshay Rao were supported by funding from the European Research Council under the European Union's Seventh Framework Programme (FP/2007–2013)/ERC Grant Agreement n. [311870] to Friederike Range. Martina Lazzaroni was supported by the Doctoral Fellowship Programme of the Austrian Academy of Sciences. Lara Bernasconi was supported by the 'Fonds Wüthrich et Mathey-Dupraz' and by the University of Neuchâtel. The funders had no role in study design, data collection and analysis, decision to publish, or preparation of the manuscript.

### Grant Disclosures

The following grant information was disclosed by the authors:
European Research Council under the European Union's Seventh Framework Programme (FP/2007–2013)/ERC Grant Agreement [311870].

Doctoral Fellowship Programme of the Austrian Academy of Sciences.
'Fonds Wüthrich et Mathey-Dupraz' by the University of Neuchâtel.

## Competing Interests

The authors declare that they have no competing interests.

## Author Contributions

- Akshay Rao conceived and designed the experiments, performed the experiments, analysed the data, prepared figures and/or tables, authored or reviewed drafts of the paper, approved the final draft.
- Lara Bernasconi conceived and designed the experiments, performed the experiments, authored or reviewed drafts of the paper, approved the final draft.
- Martina Lazzaroni conceived and designed the experiments, approved the final draft, coded videos for interobserver reliability.
- Sarah Marshall-Pescini conceived and designed the experiments, authored or reviewed drafts of the paper, approved the final draft.
- Friederike Range conceived and designed the experiments, contributed reagents/materials/analysis tools, authored or reviewed drafts of the paper, approved the final draft.

## Animal Ethics

The following information was supplied relating to ethical approvals (i.e. approving body and any reference numbers):

Special permission to use animals (wolves) in such cognitive studies is not required in Austria (Tierversuchsgesetz 2012—TVG 2012). The 'Tierversuchskommission am Bundesministerium für Wissenschaft und Forschung (Austria)' allows research without special permissions regarding animals. We obtained ethical approval for this study from the 'Ethik und Tierschutzcommission' of the University of Veterinary Medicine (Protocol number ETK-07/08/2016).

## Data Availability

The raw data are provided in the Supplemental Files.

## Supplemental Information

Supplemental information for this article can be found online at http://dx.doi.org/10.7717/peerj.5944#supplemental-information.

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
