# Peer review of "Differences in persistence between dogs and wolves in an unsolvable task in the absence of humans"

_PeerJ, doi:10.7717/peerj.5944_

## Round 0.1 · original submission · Major Revisions

I agree with the two expert reviewers that there is much of value in your manuscript. We all found the paper well-written and the experimental design appropriate and clever. The background literature appears carefully reviewed and the experiment well-controlled. Your experimental set-up allows you to tease apart ecological factors from ontogenetic experiences in a way that is unique and important. Thus, I commend you for embarking on this very timely program of research. However, Reviewer 2 expresses some concerns with the statistical approach (I will email you his R file separately as it cannot be uploaded in the system). Both reviewers indicate some places where clarification is needed. Thus, I would like to invite you to submit a revised version of the paper.
To address one of Reviewer 2’s comments, you might rephrase hypothesis 1 to ask whether dogs and wolves differ in persistence when rearing history and exposure to humans is controlled.

It would be especially nice if you also had groups of both dogs and wolves that were not socialized by humans.

Could you indicate the species or subspecies of wolves tested here?
How long did you wait before removing the object after a period of inactivity? It would be better to control the length of time each subject had access to the object. At least be very specific about the decision rule used to truncate the trials.

I’m not sure I agree that Nuru’s data should be excluded for being an outlier if there is no external reason for his increased persistence.
If the pipe test could be solved by one individual, it should not be described as an unsolvable task. There are not enough details in the methods to understand exactly what the task was.

If you wanted to present the ball first to correspond with an earlier study you still could have re-presented the ball after the pipe to test for habituation over time.

Could you have also measured the number of types of behaviors directed toward the objects as a measure of flexibility?

Minor Points:
Place a comma after approach on line 389.
Line 31 “were reinforced”
Lines 36 and 142, change “Since” to “Because”. Do not use “since” or “while” unless referring to time.
Line 38, change “which” to “that.”
Remove parentheses on lines 52-53.
On line 233, move “only” to after “with.”
Animal Cognition (2018) 21:379–392
https://doi.org/10.1007/s10071-018-1174-2
You might cite:
Johnson-Ulrich et al. Animal Cognition (2018) 21:379–392 https://doi.org/10.1007/s10071-018-1174-2, which also deals with the correlates between aspects of behavioral flexibility and persistence.

Reviewer 1 ·

Basic reporting

Line 110-116 In this sentence the authors claim that it is possible that dogs do not turn to humans to seek assistance with a problem solving task, but rather as a result of lack of persistence and that there is nothing else for the dog or wolf to do. Can they not also choose to explore the room/enclosure or simply do nothing? Persson et al. 2015 did a principal component analysis including both human-directed behaviours as well as task interactions. They showed that there was no correlation between task interactions and human interactions. If the dogs turned to the human as a result of lack of persistence, this would have shown as a negative correlation in the principal component loadings.

Line 163-164 “Based on literature, we predicted a positive correlation between persistence and behavioural variety.” ; it would be helpful if the authors could add a few of those literature references here.

Line 168 “subjects that show an unsure body posture” ; do the authors mean insecure body posture? Or do they mean that they are unsure about the body posture displayed by the individual? I think this needs clarification or perhaps just a change of word choice.

The authors write “based on literature” or similar on line 163, 172 and 175; I suggest that they re-phrase some of these sentences to make it easier and nicer to read.

Line 178-180 – I like that they summarise their aims in the end of the introduction. Very helpful!

Materials and methods

The authors do not mention the breed/s of the dogs. They refer to another study where there is a more thorough description of the animals. However, I believe that this manuscript would benefit of a bit more information about the animals such as breed, handling, social experiences etc. Have both of their species been subjects of the same studies previously e.i. could they have different experiences from previous studies possibly affecting problem-solving behaviour?

There is quite a large age difference between the dogs and the wolves of this study. This should be addressed and the implications discussed in the discussion section.

The authors use a lot of “we” e.g. “we tested”, “we placed” etc. throughout the entire materials and methods section. Hence, I believe that some re-phrasing to remove some of the “we” will improve the text in this section. Example: they write on line 212 “We mounted a smartphone..” , instead they could write “ A smartphone was mounted..” or similarly.

Line 240 I cannot find this supplementary video among the reviewing material.
Line 242 I suppose it should read “first contact with the object” and not “First contact object”?
Line 242 change “Defined” to “defined” with a small d.

Results

Line 322, 324-235, 338 etc and in Table 3 as well as in the text of the results section (perhaps also in other places) the authors write “unsure” as opposed to “confident”. I believe they are referring to the “insecure” body-posture of table 2? This should be changed so that they consistently stick with the same term. I personally think that “insecure” is preferable in this case.

Discussion

As previously mentioned, some discussion about the age difference and effects of age on problem solving behaviour should be added to this section.

Line 434-435 add a “to” “may help to better understand”

Table 1
I think this table would look better if you removed the horizontal line except for the header row. You could probably remove the vertical lines as well and make the headers and columns centered except for the first column.

Table 2
I think the design of this table can be improved, see comment on table1

Table 3
I think the design of this table can be improved, see comment on table1 e.g. remove the lines among the values but keep header lines and first column vertical line.

Figure 3
One of the labels read “unsure”, does this refer to insecure?

Figure 4
Looks good!

Figure 5
Looks good!

Experimental design

The dogs have a mean age of 4 years while the wolves had a mean age of 6.3 years. By looking at the age presented in table 1 you can see that there is a great variation where there are several dogs at the age of 2 years and several wolves at the age of 8 years. This could have effects on your data. There are studies showing that age affects boldness and explorative behaviours in dogs e.g.

Starling MJ, Branson N, Thomson PC, McGreevy PD (2013) Age, sex and reproductive status affect boldness in dogs. Vet J197: 868–872.

Siwak CT (2001) Effect of
age and level of cognitive function on spontaneous and exploratory behaviors in the beagle dog. Learn Mem 8: 317–325.

I cannot see that the authors have included age in their models. Did they correct for age in any other way? Otherwise, this needs to be done.

The authors seems to have controlled for hunger and/or food motivation in the individuals by testing them at roughly the same time of day and by not feeding them in the evening prior to testing. Even so, could it not be the case that what is tested is not really task persistence but rather food motivation or willingness to work for food. Could it be the case that the wolves in this case, found the bait more favorable than the dogs? Could this be controlled for in any way? Or, at least, it should be discussed in the discussion section.

Validity of the findings

no comment

Additional comments

In this study, the authors have investigated the difference in problem-task persistence in socialised dogs and wolves raised under similar conditions. This is important since it can give an alternative explanation to why dogs and wolves differ in their problem solving behaviour where dogs are typically less persistent and instead choose to interact with a nearby human. The authors have used a unique setup of both dogs and wolves with similar experiences of human contact resulting in a very interesting study. Generally the manuscript is well written and constructed and the findings are meaningful.

·

Basic reporting

In general, the manuscript is well-written but there are some sections that need to be rewritten for clarification. I have included several comments into my corrected pdf file.

The literature is well covered but see some of my comments for improvement.

It seems that one animal has been removed from the raw data (.xlsx file). See my comment. This might explain why I was not able to replicate some of their results using the same analyses.

I am not certain all figures should be included. For instance, fig 3 is not easy to read. Please remove or modify.

Experimental design

The experimental design is well thought and original. Excluding humans from the equation to determine whether dogs and wolves differ is very clever (I should have figured it out myself...). However, I have some problems to link the notions of persistence and flexibility to this latter. I see these questions as an afterthought.

For myself, I would have preferred that the authors had focused on testing the difference between both species first, and then, had examined the factors that could contribute to the differences, if dogs and wolves differ. The way the manuscript is written, the factors that could explain why dogs and wolves differ are examined first before testing the difference between dogs and wolves and this is not in link with the premium questions asked by the authors.

I would appreciate if more details were provided when describing the different behaviours measured by the authors and why they selected these particular behaviours. Also, as mentioned in my comments, I wonder if the behaviours were mutually exclusive one from each other. See lick and touch/lick as an example.

Validity of the findings

Having access to the dataset, I have attempted to replicate some of the results presented by the authors. Overall, I was not able to replicate the analyses.

1) I suggest to remove the two-steps cluster analysis. As described in my R file, this type of analysis is justified when the number of observations (participants) is over 200 (see attached article). In the current study, there were 27 participants. And the analysis was performed on non-normally distributed continuous variables, violating some assumptions for this test. And some variables were discrete (not continuous - as requested by a multivariate approach), with some variables having 2 possibles values. Finally, I think that a discriminant analysis would have been more useful for finding the variables that can be used to discriminate dogs and wolves. It seems to me that the two-steps cluster analysis did not answer the main question asked by the authors but it was rather used to explore the "concepts" that could be used to discriminate dogs and wolves. Just for fun, I have used R to run a two-steps cluster analysis (see R file). I was not able to replicate their results (not surprising with these variables and a small number of participants).

2) I don't understand the rational for selecting some variables in the analyses and excluding some others. Please be transparent here. See attached pdf file.

3) Some variables are discrete (e.g. behavioural variety) but appeared to be treated as continuous. These variables should be fitted using a discrete probability mass function (e.g. binomial) and not a continuous density function. Some variables (e.g. approach posture or likelihood of manipulation) are discrete. But nowhere the authors explained how they have treated them and how they were integrated into the models (e.g. for the cluster analysis).

4) I was pleased to see the use of generalized additive models for location, scale and shape. However, a better justification must be provided relative to the different distributions used to fit the data. Sometimes, instead of "playing" around with the different distributions and find the one that fit the best the data, one has to rethink about how his/her data were collected and sometimes the best option is to regroup the data or split them. Otherwise, one can face the problem of overfitting his/her data (using too much parameters).

5) Sometimes, the authors analyzed separately the two species and sometimes, they were compared within the same analysis. There is no consistency between the different analyses.

6) I was not able to replicate the Fisher test results... See my R file.

7) The rational for using some variables in different models is not presented.

8) The rational for scaling the subjects' persistence and contact latency is not provided.

9) What were the contrasts (post-hoc tests) used following the GAMLSS models?

10) I don't think that the variable "behavioural variety" is valid. My point is that some animals produced the same behaviour many many times and some animals produced the same behaviour only once. But when compiled for the behavioural variety variable, both animals were given the same score (1).

11) I did not read section 4 (Results) because of all my concerns presented above.

12) I did not read the conclusion as well.

---

## Round 0.2 · accepted · Accept

Thank you for doing such a thorough job in responding to the expert reviewers' concerns. Reviewer 2 has a few additional comments but I think these can be addressed during proofing.

# Reviewer 1 ·

Basic reporting

No comment

Experimental design

No comment

Validity of the findings

No comment

Additional comments

I am happy with how the authors have addressed my previous comments on this manuscript. Here are some minor suggestions:

Line 76: Remove an s from task-focussed to focused
Line 119 & 168: Do you really mean “teasing”?

Figure 5. The text in the upper parts of the figure is perhaps a bit too small?
Figure 6. In the generated PDF we got as reviewing material, this figure is much too small to read axis lables and legend.

Tables look good!

·

Basic reporting

no comment

Experimental design

no comment.

Validity of the findings

no comment.

Additional comments

This is my second reading of this manuscript. I just want to congratulate the authors for having modified the manuscript in lines with my previous comments, especially those related to the statistical analyses.

The conclusion is also in lines with the data and the results. More interestingly, the authors suggest several nice alternative explanations that need further investigations, such as do wolves display more flexible and adaptative behaviours than dogs.

In brief, I don't have any additional comment. Nice work.